# CAFENet: Class-Agnostic Few-Shot Edge Detection Network

## ABSTRACT

We tackle a novel few-shot learning challenge, few-shot semantic edge detection, aiming to localize boundaries of novel categories using only a few labeled samples. Reliable boundary information has been shown to boost the performance of semantic segmentation and localization, while also playing a key role in its own right in object reconstruction, image generation and medical imaging. Few-shot semantic edge detection allows recovery of accurate boundaries with just a few examples. In this work, we present a Class-Agnostic Few-shot Edge detection Network (CAFENet) based on meta-learning strategy. CAFENet employs a semantic segmentation module in small-scale to compensate for lack of semantic information in edge labels. The predicted segmentation mask is used to generate an attention map to highlight the target object region, and make the decoder module concentrate on that region. We also propose a new regularization method based on multi-split matching. In meta-training, the metric-learning problem with high-dimensional vectors are divided into smaller subproblems with low-dimensional sub-vectors. Since there are no existing datasets for few-shot semantic edge detection, we construct two new datasets, FSE-1000 and SBD-$5^i$, and evaluate the performance of the proposed CAFENet on them. Extensive simulation results confirm that the proposed CAFENet achieves better performance compared to the baseline methods using fine-tuning or few-shot segmentation.

## 1 INTRODUCTION

Semantic edge detection aims to identify pixels that belong to boundaries of predefined categories. Boundary information has been shown to be effective for boosting the performance of semantic segmentation (Bertasius et al., 2016; Chen et al., 2016) and localization (Yu et al., 2018a; Wang et al., 2015). It also plays a key role in applications such as object reconstruction (Ferrari et al., 2007; Zhu et al., 2018), image generation (Isola et al., 2017; Wang et al., 2018) and medical imaging (Abbass & Mousa, 2017; Mehena, 2019). Early edge detection algorithms interpret the problem as a low-level grouping problem exploiting hand-crafted features and local information (Canny, 1986; Sugihara, 1986). Recently, there have been significant improvements on edge detection thanks to the advances in deep learning. Moreover, beyond previous boundary detection, category-aware semantic edge detection became possible (Acuna et al., 2019; Hu et al., 2019; Yu et al., 2018b). However, it is impossible to train deep neural networks without massive amounts of annotated data.

To overcome the data scarcity issue in image classification, few-shot learning has been actively discussed for recent years (Finn et al., 2017; Lifchitz et al., 2019). Few-shot learning algorithms train machines to learn previously unseen classification tasks using only a few relevant labeled examples. More recently, the idea of few-shot learning is applied to computer vision tasks requiring highly laborious and expensive data labeling such as semantic segmentation (Dong & Xing, 2018; Wang et al., 2019) and object detection (Fu et al., 2019; Karlinsky et al., 2019). Based on meta-learning across varying tasks, the machines can adapt to unencountered environments and demonstrate robust performance in various computer vision problems. In this paper, we consider a novel few-shot learning challenge, few-shot semantic edge detection, to detect the semantic boundaries using only a few labeled samples. Through experiments, we show that few-shot semantic edge detection can not be simply solved by fine-tuning a pretrained semantic edge detector or utilizing a nonparametric edge detector in a few-shot segmentation setting. To tackle this elusive challenge, we propose a class-agnostic few-shot edge detector (CAFENet) and present new datasets for evaluating few-shot semantic edge detection.

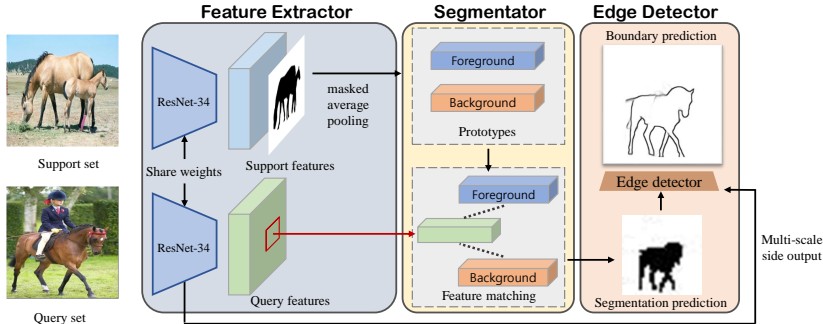

Figure 1: Architecture overview of the proposed CAFENet. The feature extractor or encoder extracts feature from the image, the segmentator generates a segmentation mask based on metric learning, and the edge detector detects semantic boundaries using the segmentation mask and query features.

Fig. 1 shows the architecture of the proposed CAFENet. Since the edge labels do not contain enough semantic information due to the sparsity of labels, performance of the edge detector severely degrades when the training dataset is very small. To overcome this, we adopt the segmentation process in advance of detecting edge with downsized feature and segmentation labels generated from boundaries labels. We utilize a simple metric-based segmentator generating a segmentation mask through pixel-wise feature matching with class prototypes, which are computed by masked average pooling of (Zhang et al., 2018). The predicted segmentation mask provides the semantic information to the edge detector. The multi-scale attention maps are generated from the segmentation mask, and applied to corresponding multi-scale features. The edge detector predicts the semantic boundaries using the attended features. Using this attention mechanism, the edge detector can focus on relevant regions while alleviating the noise effect of external details. For meta-training of CAFENet, we introduce a simple yet powerful regularization method, Multi-Split Matching Regularization (MSMR), performing metric learning on multiple low-dimensional embedding sub-spaces during meta-training.

The main contributions of this paper are as follows. First, we introduce a few-shot semantic edge detection problem for performing semantic edge detection on previously unseen objects using only a few training examples. Second, we introduce two new datasets of SBD-$5^i$ and FSE-1000 for few-shot edge detection. Third, we propose a few-shot edge detector, CAFENet and validate the performance of the proposed method through experiments.

## 2 RELATED WORK

### 2.1 FEW-SHOT LEARNING

To tackle the few-shot learning challenge, many methods have been proposed based on meta-learning. Optimization-based methods (Finn et al., 2017; Ravi & Larochelle, 2016) train the meta-learner which updates the parameters of the actual learner so that the learner can easily adapt to a new task within a few labeled samples. Metric-based methods (Vinyals et al., 2016; Snell et al., 2017; Yoon et al., 2019) train the feature extractor to assemble features from the same class together on the embedding space while keeping features from different classes far apart. Recent metric-based approaches propose dense classification (Hou et al., 2019; Kye et al., 2020). Dense classification trains an instance-wise classifier on pixel-wise classification loss which imposes coherent predictions over the spatial dimension and prevents overfitting as a result. Our model adopts the metric-based method for few-shot learning. Inspired by dense classification, we propose multi-split matching regularization which divides the feature vector into sub-vector splits and performs split-wise classification for regularization in meta-learning.

### 2.2 FEW-SHOT SEMANTIC SEGMENTATION

The goal of few-shot segmentation is to perform semantic segmentation within a few labeled samples based on meta-learning (Shaban et al., 2017; Dong & Xing, 2018; Wang et al., 2019). OSLSM of (Shaban et al., 2017) adopts a two-branch structure: conditioning branch generating element-wise scale and shift factors using the support set and segmentation branch performing segmentation with a fully convolutional network and task-conditioned features. Co-FCN (Rakelly et al., 2018) also utilizes a two-branch structure. The globally pooled prediction is generated using support set in

conditioning branch, and fused with query features to predict the mask in segmentation branch. SG-One of (Zhang et al., 2018) proposes a masked average pooling to compute prototypes from pixels of support features. The cosine similarity scores are computed between the prototypes and pixels of query feature, and the similarity map guides the segmentation process. CANet of (Zhang et al., 2019) also adopts masked average pooling to generate the global feature vector, and concatenate it with every location of the query feature for dense comparison in predicting the segmentation mask. PANet of (Wang et al., 2019) introduces prototype alignment, predicting the segmentation mask of support samples using query prediction results as labels of query samples, for regularization. PMM of Yang et al. (2020) utilizes multiple prototypes generated using Expectation-Maximization (EM) algorithm to effectively leverage the semantic information from the few labeled samples.

## 2.3 SEMANTIC EDGE DETECTION

Semantic edge detection aims to find the boundaries of objects from image and classify the objects at the same time. The history of semantic edge detection (Acuna et al., 2019; Hu et al., 2019) dates back to the work of (Prasad et al., 2006) which adopts the support vector machine as a semantic classifier on top of the traditional canny edge detector. Recently, many semantic edge detection algorithms rely on deep neural network and multi-scale feature fusion. CASENET of (Yu et al., 2017) addresses the semantic edge detection as a multi-label problem where each boundary pixel is labeled into categories of adjacent objects. Dynamic Feature Fusion (DFF) of (Hu et al., 2019) proposes a novel way to leverage multi-scale features. The multi-scale features are fused by weighted summation with fusion weights generated dynamically for each images and each pixel. Meanwhile, Simultaneous Edge Alignment and Learning (SEAL) of (Yu et al., 2018b) deals with severe annotation noise of the existing edge dataset (Hariharan et al., 2011). SEAL treats edge labels as latent variables and jointly trains them to align noisy misaligned boundary annotations. Semantically Thinned Edge Alignment Learning (STEAL) of (Acuna et al., 2019) improves the computation efficiency of edge label alignment through a lightweight level set formulation.

## 3 PROBLEM SETUP

For few-shot semantic edge detection, we use train set $D_{train}$ and test set $D_{test}$ consisting of non-overlapping categories $C_{train}$ and $C_{test}$. The model is trained only using $C_{train}$, and the test categories $C_{test}$ are never seen during the training phase. For meta-training of the model, we adopt episodic training as done in many previous few-shot learning works. Each episode is composed of a support set with a few-labeled samples and a query set. When an episode is given, the model adapts to the given episode using the support set and detect semantic boundaries of the query set. By episodic training, the model is learned so that it adapts to the unseen class using only a few labeled samples and predict semantic edges of query samples.

For $N_c$-way $N_s$-shot setting, each training episode is constructed by $N_c$ classes sampled from $C_{train}$. When $N_c$ categories are given, $N_s$ support samples and $N_q$ query samples are randomly chosen from $D_{train}$ for each class. In evaluation, the performance of the model is evaluated using test episodes. The test episodes are constructed in the same way as the training episodes, except $N_c$ classes and corresponding support and query samples are sampled from $C_{test}$ and $D_{test}$.

In this work, we address $N_c$-way $N_s$-shot semantic edge detection. The goal is training the model to generalize to $N_c$ unseen classes given only $N_s$ images and their edge labels. Based on the few labeled support samples, the model should produce edge predictions of query images which belong to $N_c$ unencountered classes.

## 4 METHOD

We propose a novel algorithm for few-shot semantic edge detection. Fig. 2 illustrates the network architecture of the proposed method. The proposed CAFENet adopts the semantic segmentation module to compensate for the lack of semantic information in edge labels. The predicted segmentation mask is utilized for attention in skip connection. The final edge detection is done using attentive multi-scale features.

## 4.1 SEMANTIC SEGMENTATOR

Most previous works on semantic edge detection directly predict edges from the given input image. However, direct edge prediction is a hard task when only a few labeled samples are given. To overcome this difficulty in few-shot edge detection, we adopt a semantic segmentation module in advance of edge prediction. With the assistance of the segmentation module, CAFENet can effectively

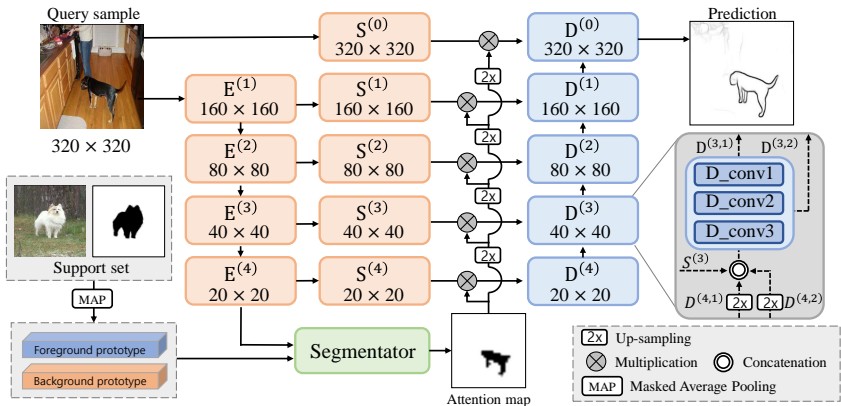

Figure 2: Network architecture overview of proposed CAFENet. ResNet-34 encoder $E^{(1)} \sim E^{(4)}$ extracts multi-level semantic features. The segmentator module generates a segmentation prediction using query feature from $E^{(4)}$ and prototypes $P_{FG}, P_{BG}$ from support set features. Small bottleneck blocks $S^{(0)} \sim S^{(4)}$ transform the original image and multi-scale features from encoder blocks to be more suitable for edge detection. The attention maps generated from segmentation prediction are applied to multi-scale features to localize the semantically related region. Decoder $D^{(0)} \sim D^{(4)}$ takes attentive multi-scale features to give edge prediction.

localize the target object and extract semantic features from query samples. For few-shot segmentation, we employ the metric-learning which utilizes prototypes for foreground and background as done in (Dong & Xing, 2018; Wang et al., 2019). Given the support set $S = \{x_i^s, y_i^s\}_{i=1}^{N_s}$, the encoder $E$ extracts features $\{E(x_i^s)\}_{i=1}^{N_s}$ from $S$. Also, for support labels $\{y_i^s\}_{i=1}^{N_s}$, we generate the dense segmentation mask $\{M_i^s\}_{i=1}^{N_s}$ using a rule-based preprocessor, considering the pixels inside the boundary as foreground pixels in the segmentation label. Using down-sampled segmentation labels $\{m_i^s\}_{i=1}^{N_s}$, the prototype for foreground pixels $P_{FG}$ is computed as

$$P_{FG} = \frac{1}{N_s} \frac{1}{H \times W} \sum_i \sum_j E_j(x_i^s) m_{i,j}^s \qquad (1)$$

where $j$ indexes the pixel location, $E_j(x)$ and $m_{i,j}^s$ denote the $j$th pixel of feature $E(x)$ and segmentation mask $m_i^s$. $H, W$ denote height and width of the images. Likewise, the background prototype $P_{BG}$ is computed as

$$P_{BG} = \frac{1}{N_s} \frac{1}{H \times W} \sum_i \sum_j E_j(x_i^s)(1 - m_{i,j}^s). \qquad (2)$$

Following the prototypical networks of (Snell et al., 2017), the probability that pixel $j$ belongs to foreground for the query sample $x_i^q$ is

$$p(y_{i,j}^q = FG|x_i^q; E) = \frac{exp(-\tau d(E_j(x_i^q), P_{FG}))}{exp(-\tau d(E_j(x_i^q), P_{FG})) + exp(-\tau d(E_j(x_i^q), P_{BG}))} \qquad (3)$$

where $d(\cdot, \cdot)$ is squared Euclidean distance between two vectors and $\tau$ is a learnable temperature parameter. With query samples $\{x_i^q\}_{i=1}^{N_q}$ and the down-sampled segmentation labels for query $\{m_i^q\}_{i=1}^{N_q}$, the segmentation loss $L_{Seg}$ is calculated as the mean-squared error (MSE) loss between predicted probabilities and the down-sized segmentation mask

$$L_{Seg} = \frac{1}{N_q} \frac{1}{H \times W} \sum_{i=1}^{N_q} \sum_{j=1}^{H \times W} \{(p(y_{i,j}^q = FG|x_i^q; E) - m_{i,j}^q)^2\}. \qquad (4)$$

Note that the segmentation mask is generated in down-sized scale so that any pixel near the boundaries can be classified into the foreground to some extent, as well as the background. Therefore, we approach the problem as a regression using MSE loss rather than cross entropy loss.

## 4.2 MULTI-SPLIT MATCHING REGULARIZATION

The metric-based few-shot segmentation method utilizes distance metrics between the high-dimensional feature vectors and prototypes, as seen in Fig. 3a. However, this approach is prone

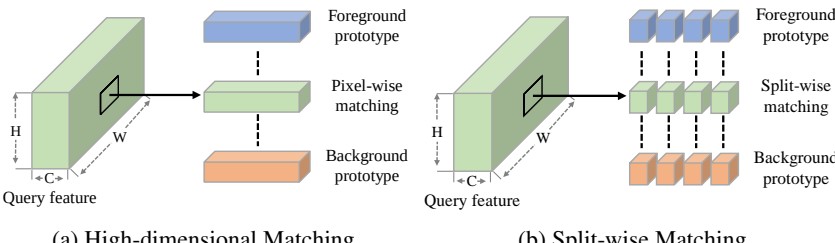

Figure 3: Comparison between (a) High-dimensional feature matching used in (Dong & Xing, 2018; Wang et al., 2019) and (b) split-wise feature matching in MSMR

to overfit due to the massive number of parameters in feature vectors. To get around this issue, we propose a novel regularization method, multi-split matching regularization (MSMR). In MSMR, high-dimensional feature vectors are split into several low-dimensional feature vectors, and the metric learning is conducted on each vector split as Fig. 3b.

With the query feature $E(x_i^q) \in \mathbb{R}^{C \times W \times H}$, where $C$ is channel dimension and $H, W$ are spatial dimensions, we divide $E(x_i^q)$ into $K$ sub-vectors $\{E^k(x_i^q)\}_{k=1}^{K}$ along channel dimension. Each sub-vector $E^k(x_i^q)$ is in $\mathbb{R}^{\frac{C}{K} \times W \times H}$. Likewise, the prototypes $P_{FG}$ and $P_{BG}$ are also disassembled into $K$ sub-vectors $\{P_{FG}^k\}_{k=1}^{K}$ and $\{P_{BG}^k\}_{k=1}^{K}$ along channel dimension where $P_{FG}^k, P_{BG}^k \in \mathbb{R}^{\frac{C}{K}}$ . For the $k^{th}$ sub-vector of query feature $E^k(x_i^q)$, the probability that the $j^{th}$ pixel belongs to the foreground class is computed as follows:

$$p^k(y_{i,j}^q = FG|x_i^q; E) = \frac{exp(-\tau d(E_j^k(x_i^q), P_{FG}^k))}{exp(-\tau d(E_j^k(x_i^q), P_{FG}^k)) + exp(-\tau d(E_j^k(x_i^q), P_{BG}^k))}. \qquad (5)$$

MSMR divides the original metric learning problem into $K$ small sub-problems composed of a fewer parameters and acts as regularizer for high-dimensional embeddings. The prediction results of $K$ sub-problems are reflected on learning by combining the split-wise segmentation losses to original segmentation loss in Eq. 4. The total segmentation loss is calculated as

$$L_{Seg} = \frac{1}{N_q} \frac{1}{H \times W} \sum_{i=1}^{N_q} \sum_{j=1}^{H \times W} \{(p_{i,j} - m_{i,j}^q)^2 + \sum_{i=1}^{K}(p_{i,j}^k - m_{i,j}^q)^2\}. \qquad (6)$$

where $p_{i,j} = p(y_{i,j}^q = FG|x_i^q; E)$ and $p_{i,j}^k = p^k(y_{i,j}^q = FG|x_i^q; E)$.

### 4.3 ATTENTIVE EDGE DETECTOR

As shown in Fig. 2, we adopt the nested encoder structure to extract rich hierarchical features. The multi-scale side outputs from encoder $E^{(1)} \sim E^{(4)}$ are post-processed through bottleneck blocks $S^{(1)} \sim S^{(4)}$. Since ResNet-34 gives side outputs of down-sized scale, we pass the original image through bottleneck block $S^{(0)}$ to extract local details in original scale. In front of $S^{(3)}$, we employ the Atrous Spatial Pyramid Pooling (ASPP) block of (Chen et al., 2017). We have empirically found that locating ASPP there shows better performance.

In utilizing multi-scale features, we employ the predicted segmentation mask $\hat{M}$ from the segmentator where the $j^{th}$ pixel of $\hat{M}$ is the predicted probability from Eq. 3. Note that we generate $\hat{M}$ based on the entire feature vectors and the prototypes instead of utilizing sub-vectors, since the split-wise metric learning is used only for regularizing the segmentation module. For each layer $l$, $\hat{M}^{(l)}$ denotes the segmentation mask upscaled to the corresponding feature size by bilinear interpolation. Using segmentation prediction mask $\hat{M}^{(l)}$, we generate attention map $A^{(l)}$, as follows. First, the prediction with a value lower than threshold $\lambda$ is rounded down to zero, to ignore activation in regions with low confidence. Second, we broaden the attention map using morphological dilation of (Feng et al., 2019) as a second chance, since the segmentation module may not always guarantee fine results. The final attention map of $l^{th}$ layer $A^{(l)}$ is computed as follows

$$A^{(l)} = \mathbb{1}(\hat{M}^{(l)} > \lambda)\hat{M}^{(l)} + Dilation(\mathbb{1}(\hat{M}^{(l)} > \lambda)\hat{M}^{(l)}) \qquad (7)$$

where $\mathbb{1}(\hat{M}^{(l)} > \lambda)\hat{M}^{(l)}$ is the rounded value of prediction mask $\hat{M}^{(l)}$. The attention maps are applied to the multi-scale features of corresponding bottleneck blocks $S^{(0)} \sim S^{(4)}$. We apply the

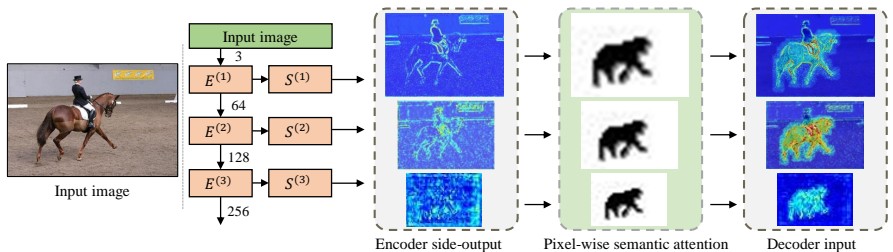

Figure 4: An example of activation map of (Yosinski et al., 2015) before and after pixel-wise semantic attention (warmer color has higher value). As seen, the attention mechanism makes encoder side-outputs attend to the regions of the target object (*horse* in the figure).

residual attention of (Hou et al., 2019), where the initial multi-level side outputs from $S^{(l)}$ are pixel-wisely weighted by $1 + A^{(l)}$, to strengthen the activation value of the semantically important region. We visualize the effect of semantic attention in Fig. 4.

As shown in Fig. 2, the decoder network is composed of five consecutive convolutional blocks. Each decoder block $D^{(l)}$ contains three $3 \times 3$ convolution layers. The outputs of decoder blocks $D^{(1)} \sim D^{(4)}$ are bilinearly upsampled by two and passed to the next block. Similar to (Feng et al., 2019), the up-sampled decoder outputs are then concatenated to the skip connection features from bottleneck blocks $S^{(0)} \sim S^{(4)}$ and previous decoder blocks. Multi-scale semantic information and local details are transmitted through skip architectures. The hierarchical decoder network in turn refines the outputs of the previous decoder blocks and finally produces the edge prediction $\hat{y}_i^q$ of query samples $x_i^q$.

Following the work of (Deng et al., 2018), we combine cross-entropy loss and Dice loss to produce crisp boundaries. Given a query set $Q = \{x_i^q, y_i^q\}_{i=1}^{N_q}$ and prediction mask $\hat{y}_i^q$, the cross-entropy loss is computed as

$$L_{CE} = -\sum_{i=1}^{N_q} \{ \sum_{j \in Y_+} log(\hat{y}_i^q) + \sum_{j \in Y_-} log(1 - \hat{y}_i^q) \} \tag{8}$$

where $Y_+$ and $Y_-$ denote the sets of foreground and background pixels. The Dice loss is then computed as

$$L_{Dice} = \sum_{i=1}^{N_q} \{ \frac{\sum_j (\hat{y}_{i,j}^q)^2 + \sum_j (y_{i,j}^q)^2}{2 \sum_j \hat{y}_{i,j}^q y_{i,j}^q} \} \tag{9}$$

where j denotes the pixels of a label. The final loss for meta-training is given by

$$L_{final} = L_{Seg} + L_{CE} + L_{Dice}. \tag{10}$$

## 5 EXPERIMENTS

### 5.1 DATASETS

#### 5.1.1 SBD-$5^i$

Based on the SBD dataset of (Hariharan et al., 2011) for semantic edge detection, we propose a new SBD-$5^i$ dataset. With reference to the setting of Pascal-$5^i$, 20 classes of the SBD dataset are divided into 4 splits. In the experiment with split $i$, 5 classes in the $i$th split are used as test classes $C_{test}$. The remaining 15 classes are utilized as training classes $C_{train}$. The training set $D_{train}$ is constructed with all image-annotation pairs whose annotation include at least one pixel from the classes in $C_{train}$. For each class, the boundary pixels which do not belong to that class are considered as background. The test set $D_{test}$ is also constructed in the same way as $D_{train}$, using $C_{test}$ this time. Considering the difficulty of few-shot setting and severe annotation noise of the SBD dataset, we extract thicker edges. We utilize edges extracted from the segmentation mask as ground truth instead of original boundary labels of the SBD dataset, and thickness of extracted edge lies between $3 \sim 4$ pixels on average. We conduct 4 experiments with each split of $i = 0 \sim 3$, and report performance of each split as well as the averaged performance. Note that unlike Pascal-$5^i$, we do not consider division of training and test samples of the original SBD dataset. As a result, the images in $D_{train}$ might appear in $D_{test}$ with different annotation from class in $C_{test}$.

### 5.1.2 FSE-1000

The datasets used in previous semantic edge detection research such as SBD of (Hariharan et al., 2011) and Cityscapes of (Cordts et al., 2016) are not suitable for few-shot learning as they have only 20 and 30 classes, respectively. We propose a new dataset for few-shot edge detection, which we call FSE-1000, based on FSS-1000 of (Wei et al., 2019). FSS-1000 is a dataset for few-shot segmentation and composed of 1000 classes and 10 images per class with foreground-background segmentation annotation. From the images and segmentation masks of FSS-1000, we build FSE-1000 by extracting boundary labels from segmentation masks. As done in SBD-$5^i$, we extract thick edges of which thickness is around $2 \sim 3$ pixels on average in the light of difficulty associated with few-shot setting. For dataset split, we split 1000 classes into 800 training classes and 200 test classes. We will provide the detailed class configuration in the Supplementary Material.

### 5.2 Evaluation Settings

We use two evaluation metrics to measure the few-shot semantic edge detection performance of our approach: the Average Precision (AP) and the maximum F-measure (MF) at optimal dataset scale (ODS). In evaluation, we compare the unthinned raw prediction results and the ground truths without Non-Maximum Suppression (NMS) following (Acuna et al., 2019; Yu et al., 2018b). For the evaluation of edge detection, an important parameter is matching distance tolerance which is an error threshold between the prediction result and the ground truth. Prior works on edge detection such as (Acuna et al., 2019; Hariharan et al., 2011; Yu et al., 2017; 2018b) adopt non-zero distance tolerance to resolve the annotation noise. However, the proposed datasets for few-shot edge detection utilize thicker boundaries to overcome the annotation noise issue instead of adopting distance tolerance. Moreover, evaluation with non-zero distance tolerance requires additional heavy computation. This becomes more problematic under few-shot setting where the performance should be measured on the same test image multiple times due to the variation in the support set. For these reasons, we set distance tolerance to be 0 for both FSE-1000 and SBD-$5^i$. In addition, we evaluate the positive predictions from the area inside an object and zero-padded region as false positives, which is stricter than the evaluation protocol in prior works of (Hariharan et al., 2011; Yu et al., 2017).

### 5.3 Implementation Detail

We implement our framework using Pytorch library and adopt Scikit-learn library to construct the precision-recall curve and compute average precision (AP). For the encoder, ResNet-34 pretrained on ImageNet is adopted. All parameters except the encoder parameters are learned from scratch. The entire network is trained using the Adam optimizer of (Kingma & Ba, 2014) with weight decay regularization of (Loshchilov & Hutter, 2017). In both experiments on FSE-1000 and SBD-$5^i$, we use a learning rate of $10^{-4}$ and an $l2$ weight decay rate of $10^{-2}$. For FSE-1000 experiments, the model is trained with 40,000 episodes and the learning rate is decayed by 0.1 after training 38,000 episodes. For SBD-$5^i$ experiments, 30,000 episodes are used for training, and the learning rate is decayed by 0.1 after training 28,000 episodes. Higher shot training of (Liu et al., 2019) is employed in 1-shot experiments for both datasets.

#### 5.3.1 Data preprocessing

During training, we adopt data augmentation with random rotation by multiples of 90 degrees for both SBD-$5^i$ and FSE-1000. We additionally resize SBD-$5^i$ data to 320×320, while no such resizing is performed on FSE-1000. During evaluation, images of SBD-$5^i$ are zero-padded to 512×512. Again, the original image size is used for FSE-1000.

### 5.4 Experiment Result

Table 1 shows the experiment results on the SBD-$5^i$ dataset. To verify the value of the proposed method, we compare CAFENet with two baselines. The first baseline is a fine-tuned edge detection model with only a few labeled samples. Meta-learning strategy is not used for the first baseline. We employ the DFF of (Hu et al., 2019), utilizing the implementation offered by the authors. For each split of SBD-$5^i$, we pretrain a 15-way semantic edge detector with training classes and fine-tune the pretrained edge detector with a few labeled samples for new classes in test split. During pretraining, we follow the training strategies and hyperparameters of (Hu et al., 2019). In fine-tuning, we randomly initialize some sub-modules that are closely related to final prediction ("side5", "side5-w", and "ada-learner") and train them altogether using the support images.

The second baseline is constructed by combining a non-parametric edge detectors such as Canny detector or Sobel detector with a few-shot segmentation algorithm. The semantic edge detection is occasionally interpreted as a dual task of semantic segmentation, but prior work of (Acuna et al., 2019) verifies that the semantic edge detector outperforms the segmentator combined with the Sobel edge detector, and demonstrates the importance of semantic edge detection. In our experiments, we combine PANet and PMM, the state-of-the-art few-shot segmentation method, with the Sobel edge detector. To experiment with existing segmentation methods, we utilize the implementation provided by the authors. For each split of SBD-$5^i$, we meta-train the PANet and PMM on training classes using the segmentation labels. In evaluation, we obtain the edge predictions of test classes by applying the Sobel edge detector on the segmentation masks as done in (Acuna et al., 2019), and compare the predictions with the edge labels of test classes.

We utilize the ResNet-34 backbone for CAFENet and PANet. For PMM, we utilize ResNet-50 backbone. We also employ higher shot training in 1-shot experiments for both baselines as done in CAFENet experiments. The results in Table 1 show that the proposed CAFENet outperforms all baselines in both MF and AP scores by significant margin. The experiment results prove that the few-shot semantic edge detector can not be simply substituted by few-shot segmentator or the fine-tuned semantic edge detector. In Table 2, the experiment results on the FSE-1000 dataset are shown. For FSE-1000, we only experiment with the few-shot segmentation baseline since it is hard to train a semantic edge detector with training set of FSE-1000, due to the large number of training classes. We can see that the proposed CAFENet outperforms the baseline even when the dataset contains more diverse classes.

Table 1: Evaluation results of proposed CAFENet on SBD-$5^i$. 1000 randomly sampled test episodes are used for evaluation. MF and AP scores are measured by %

| Metric | Method (5-shot) | SBD-$5^0$ | SBD-$5^1$ | SBD-$5^2$ | SBD-$5^3$ | Mean |
|---|---|---|---|---|---|---|
| MF (ODS) | DFF + Finetune | 9.25 | 8.63 | 8.08 | 7.83 | 8.45 |
| | PANet + Sobel | 19.66 | 23.48 | 21.10 | 18.09 | 20.58 |
| | PMM + Sobel | 31.73 | 29.99 | 29.91 | 26.03 | 29.42 |
| | CAFENet (Ours) | **34.71** | **36.81** | **32.02** | **28.37** | **32.98** |
| AP | DFF + Finetune | 9.17 | 6.77 | 7.04 | 6.14 | 7.28 |
| | PANet + Sobel | 11.91 | 14.13 | 11.92 | 9.42 | 11.85 |
| | PMM + Sobel | 23.26 | 20.76 | 20.38 | 17.94 | 20.59 |
| | CAFENet (Ours) | **30.47** | **32.40** | **27.01** | **23.06** | **28.24** |
| Metric | Method (1-shot) | SBD-$5^0$ | SBD-$5^1$ | SBD-$5^2$ | SBD-$5^3$ | Mean |
| MF (ODS) | DFF + Finetune | 8.87 | 5.54 | 4.91 | 2.95 | 5.57 |
| | PANet + Sobel | 19.33 | 23.01 | 20.79 | 17.62 | 20.19 |
| | PMM + Sobel | 31.18 | 29.23 | 29.38 | 25.65 | 28.86 |
| | CAFENet (Ours) | **31.54** | **34.75** | **29.47** | **26.68** | **30.61** |
| AP | DFF + Finetune | 7.91 | 3.71 | 3.31 | 1.55 | 4.12 |
| | PANet + Sobel | 12.32 | 14.39 | 12.13 | 9.76 | 12.15 |
| | PMM + Sobel | 22.77 | 20.21 | 19.85 | 17.56 | 20.10 |
| | CAFENet (Ours) | **26.81** | **29.08** | **23.77** | **20.44** | **25.03** |

Table 2: 1-way 1-shot and 1-way 5-shot results of proposed CAFENet on FSE-1000. 1000 randomly sampled test episodes are used for evaluation. MF and AP scores are measured by %

| Metric | Method | 1-way 1-shot | 1-way 5-shot |
|---|---|---|---|
| MF (ODS) | PANet + Sobel | 38.38 | 38.78 |
| | PMM + Sobel | 36.31 | 36.46 |
| | CAFENet (Ours) | **58.47** | **60.63** |
| AP | PANet + Sobel | 27.85 | 28.11 |
| | PMM + Sobel | 31.22 | 31.82 |
| | CAFENet (Ours) | **60.54** | **63.92** |

## 6 CONCLUSION

In this paper, we establish the few-shot semantic edge detection problem. We proposed the Class-Agnostic Few-shot Edge detector (CAFENet) based on a skip architecture utilizing multi-scale features. To compensate the shortage of semantic information in edge labels, CAFENet employs a segmentation module in low resolution and utilizes segmentation masks to generate attention maps. The attention maps are applied to multi-scale skip connection to localize the semantically related region. We also present the MSMR regularization method splitting the feature vectors and prototypes into several low-dimension sub-vectors and solving multiple metric-learning sub-problems with the sub-vectors. We built two novel datasets of FSE-1000 and SBD-$5^i$ well-suited to few-shot semantic edge detection. Experimental results demonstrate that the proposed method significantly outperforms the baseline approaches relying on fine-tuning or few-shot semantic segmentation.

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

## A  ADDITIONAL EXPERIMENTAL SETUP

In this section, we provide detailed information about experimental setup. We adopt the ImageNet pretrained ResNet-34 with 64-128-256-512 channels for each residual block from [pytorch framework] as the encoder. To construct the skip architecture, we employ the bottleneck block of ResNet as the post-processing blocks $S^{(1)} \sim S^{(4)}$. Each bottleneck block consists of two 1x1 convolutional layers and one 3x3 convolutional layer with expansion rate of 4. Dropout with the ratio of 0.25 is applied to the end of each bottleneck block. For the ASPP Module in front of $S^{(3)}$, we adopt the dilation rate of 1,4,7,11. The segmentation module generates a segmentation prediction with the rounding threshold value $\lambda$ of 0.4. For decoder, each decoder block is composed of three consecutive 3x3 convolutional layers, and dropout with the ratio of 0.25 is again located at the end of each layer.

During meta-training of CAFENet, we set the number of query samples in training episodes to be 5 for FSE-1000 and 10 for SBD-$5^i$, respectively. In evaluation, we employ *average_precision_score* function of Scikit-learn library to measure the Average Precision (AP) score. We compute the AP score for each image and average them to measure the overall performance. For Maximum F-measure (MF) score, we measure true positives (TP), false positives (FP) and false negatives (FN) at every 0.01 threshold intervals for each image, and accumulate the values for all images in 1000 test episodes. The MF score is computed using the accumulated TP, FP, and FN values.

# B  ABLATION STUDIES

In this section, we shot the results of ablation experiments to examine the impact of proposed MSMR and attentive decoder. Table B.1 shows the experiment results on the FSE-1000 dataset. The baseline method in Table B.1 conducts edge prediction in low resolution and utilizes the auxiliary loss from low resolution together with the loss from the edge prediction in original resolution for meta-training. The edge prediction is done using a metric-based method utilizing prototypes which are computed using down-sampled edge labels. The method dubbed as **Seg** utilizes a segmentation module without MSMR or attentive decoding. **Seg** directly matches high-dimensional query feature vectors with prototypes in both training and evaluation. In **Seg**, the segmentation module is utilized only to provide the segmentation loss that assists for model learning to extract semantic features. **Seg + Att** employs the predicted segmentation mask for the additional attention process in skip architecture. **Seg + MSMR + Att** additionally utilizes the MSMR regularization for training. For fair comparison, all methods use the same network architecture and training hyperparameters. For SBD-$5^i$ datasets, the ablation experiments are done with same model variations as FSE-1000. The results on SBD-$5^i$ are shown in Table B.2.

Table B.1: Ablation experiment results of proposed CAFENet on FSE-1000. 1000 randomly sampled test episodes are used for evaluation. MF and AP scores are measured by %

| Metric | Method | 1-way 1-shot | 1-way 5-shot |
|---|---|---|---|
| MF (ODS) | baseline | 52.71 | 53.52 |
| | Seg | 56.89 | 59.65 |
| | Seg + Att | 58.00 | 60.14 |
| | Seg + Att + MSMR | **58.47** | **60.63** |
| AP | baseline | 53.66 | 54.59 |
| | Seg | 58.80 | 61.87 |
| | Seg + Att | 59.81 | 62.37 |
| | Seg + Att + MSMR | **60.54** | **63.92** |

Tables B.1 and B.2 demonstrate that the use of the segmentation module in **Seg** gives significant performance advantages over baseline for both FSE-1000 and SBD-$5^i$ datasets. It is also seen that the additional use of attentive decoding, **Seg + Att**, generally improves the performance over **Seg**. Finally, adding the effect of MSMR regularization gives substantial extra gains, as seen by the scores associated with **Seg + MSMR + Att**. Clearly, when compared with baseline, our overall approach **Seg + MSMR + Att** provides large gains. In the main paper, we report the results of **Seg + MSMR + Att** as the results of CAFENet.

## B.1  ADDITIONAL EXPERIMENTS ON MULTI-SPLIT MATCHING REGULARIZATION

### B.1.1  FEATURE MATCHING METHOD FOR SEGMENTATION

In Table B.3, we have compared various feature matching methods between prototypes and query feature vectors for producing segmentation prediction on SBD-$5^i$. The method **baseline** refers to the original method generating segmentation prediction using only the similarity metric between high-dimensional vectors as done in Eq. 3. For the method **average**, segmentation predictions from low-dimensional feature splits (Eq. 5) and original high-dimensional feature vectors (Eq. 5) are averaged to generate the final prediction mask. The **average** method can be understood as a method

Table B.2: Ablation experiment results of proposed CAFENet on SBD-$5^i$. 1000 randomly sampled test episodes are used for evaluation. MF and AP scores are measured by %

| | i=0 | i=1 | | i=2 | | i=3 | |
|---|---|---|---|---|---|---|---|
| | aeroplane,bike,bird,boat,bottle | bus,car,cat,chair,cow | | table,dog,horse,mbike,person | | plant,sheep,sofa,train,tv | |
| Metric | Method (5-shot) | SBD-$5^0$ | SBD-$5^1$ | SBD-$5^2$ | SBD-$5^3$ | Mean | |
| MF (ODS) | baseline | 22.27 | 19.64 | 20.41 | 20.41 | 20.20 | |
| | Seg | 30.61 | 31.62 | 28.06 | 24.97 | 28.82 | |
| | Seg + Att | 31.75 | 33.41 | 28.44 | 26.03 | 29.91 | |
| | Seg + Att + MSMR | **34.71** | **36.81** | **32.02** | **28.37** | **32.98** | |
| AP | baseline | 18.68 | 15.57 | 14.97 | 14.05 | 15.82 | |
| | Seg | 26.14 | 26.78 | 21.92 | 18.43 | 23.32 | |
| | Seg + Att | 27.61 | 28.39 | 22.66 | 20.11 | 24.69 | |
| | Seg + Att + MSMR | **30.47** | **32.40** | **27.01** | **23.06** | **28.24** | |
| Metric | Method (1-shot) | SBD-$5^0$ | SBD-$5^1$ | SBD-$5^2$ | SBD-$5^3$ | Mean | |
| MF (ODS) | baseline | 21.81 | 19.49 | 20.34 | 18.06 | 19.93 | |
| | Seg | 29.89 | 31.64 | 27.89 | 24.41 | 28.46 | |
| | Seg + Att | 30.72 | 33.03 | 28.63 | 25.04 | 29.36 | |
| | Seg + Att + MSMR | **31.54** | **34.75** | **29.47** | **26.68** | **30.61** | |
| AP | baseline | 18.11 | 15.47 | 14.73 | 13.89 | 15.55 | |
| | Seg | 25.16 | 26.15 | 21.52 | 18.52 | 22.84 | |
| | Seg + Att | 26.10 | 27.21 | 22.47 | 18.81 | 23.65 | |
| | Seg + Att + MSMR | **26.81** | **29.08** | **23.77** | **20.44** | **25.03** | |

utilizing MSMR not only for regularization, but also for inference. In the **weighted sum** method, the above five segmentation masks are combined using a weighted sum with learnable weights. As we can see in Table B.3, the MSMR method shows the best performance when employed for regularization.

Table B.3: Comparison of different feature matching method on SBD-$5^i$ under 1-way 5-shot setting. MF and AP scores are averaged over 4 splits

| Feature matching method | baseline | average | weighted sum |
|---|---|---|---|
| AP | **34.61** | 31.05 | 31.44 |
| MF(ODS) | **29.91** | 26.20 | 26.46 |

### B.1.2 NUMBER OF VECTOR SPLITS

MSMR divides the high-dimensional feature into multiple splits. Table B.4 shows the performance of proposed CAFENet with varying numbers of splits $K$. Comparing the $K = 1$ case with other cases, we can see that applying MSMR regularization consistently improves performance. We can see that $K = 4$ results in the best AP and MF performance. The performance gain is marginal when we divide the embedding dimension into too small ($K = 16$) or too big ($K = 2$) a pieces.

Table B.4: Comparison of different numbers of vector splits $K$ on SBD-$5^i$ under 1-way 5-shot setting. MF and AP scores are averaged over 4 splits

| Number of splits | $K = 1$ | $K = 2$ | $K = 4$ | $K = 8$ | $K = 16$ |
|---|---|---|---|---|---|
| AP | 24.69 | 26.48 | **29.91** | 27.68 | 23.83 |
| MF(ODS) | 29.91 | 31.62 | **32.30** | 31.58 | 30.77 |

### B.1.3 RANDOM PROJECTION FOR VECTOR SPLITS

In MSMR, we divide query feature into K splits along the channel dimension, i.e. we apply deterministic splitting. In Table B.5, we compare the performance of different splitting methods. The method dubbed as **Deterministic** refers to the MSMR method that we utilize in CAFENet. The **Random** method randomly splits feature vectors into 4 parts in each episode. The **Baseline** method does not split features at all. Interestingly, **Random**'s performance significantly degrades, even below that of **Baseline**.

Table B.5: Comparison of different splitting methods on SBD-$5^i$. MF and AP scores are measured by %

| Metric | Method (5-shot) | SBD-$5^0$ | SBD-$5^1$ | SBD-$5^2$ | SBD-$5^3$ | Mean |
|---|---|---|---|---|---|---|
| MF (ODS) | Deterministic | **34.71** | **36.81** | **32.02** | **28.37** | **32.98** |
| | Baseline | 31.75 | 33.41 | 28.44 | 26.03 | 29.91 |
| | Random | 23.67 | 17.22 | 21.16 | 20.51 | 20.64 |
| AP | Deterministic | **30.47** | **32.40** | **27.01** | **23.06** | **28.24** |
| | Baseline | 27.61 | 28.39 | 22.66 | 20.11 | 24.69 |
| | Random | 20.37 | 17.51 | 18.26 | 15.76 | 18.07 |
| Metric | Method (1-shot) | SBD-$5^0$ | SBD-$5^1$ | SBD-$5^2$ | SBD-$5^3$ | Mean |
| MF (ODS) | Deterministic | **31.54** | **34.75** | **29.47** | **26.68** | **30.61** |
| | Baseline | 30.72 | 33.03 | 28.63 | 25.04 | 29.36 |
| | Random | 23.36 | 17.04 | 21.43 | 20.47 | 20.58 |
| AP | Deterministic | **26.81** | **29.08** | **23.77** | **20.44** | **25.03** |
| | Baseline | 26.10 | 27.21 | 22.47 | 18.81 | 23.65 |
| | Random | 20.12 | 17.35 | 18.57 | 15.64 | 17.92 |

## B.2 ADDITIONAL EXPERIMENTS ON SEMANTIC ATTENTION

### B.2.1 EXPERIMENTS ON APPLYING THE SEMANTIC ATTENTION

In the proposed CAFENet, we utilize the semantic attention in the attentive decoder. In Table B.6, we compare three methods that utilize the attention in different manners. **Attentive Decoding** is our proposed CAFENet that applies the semantic attention to the multi-scale features. The second method, **Direct Attention**, is a method that directly passes the feature to the edge detector and applies the semantic attention to the final edge prediction. The last method, **No Attention** is a baseline where the edge detector generates prediction without any attention. For both 1-shot and 5-shot settings, regardless of how the attention is applied, semantic attention considerably improves performance. These results show the effectiveness of semantic attention. The results also show that the proposed approach of **Attentive Decoding** yields better results compared to **Direct Attention**.

### B.2.2 EXPERIMENTS ON GENERATING THE ATTENTION

In the proposed attentive decoding, we can generate an attention map using various methods. In Table.B.7, we compare two different methods to generate the attention map. **Segmentation Attention** is the method adopted in proposed CAFENet. In **Segmentation Attention**, the output of the segmentation module $S$ is utilized as the $\hat{M}$ in equation 7. In **Edge Attention**, the edge prediction $E(S)$ is generated from segmentation mask $S$ by equation 11 following (Feng et al. (2019)). The generated edge prediction $E(S)$ is then used as the $\hat{M}$ in equation 7. Experiment results show that utilizing the segmentation mask to generate the attention map performs better than utilizing the edge prediction.

$$E(S) = |S - AvgPool(S)| \qquad (11)$$

Table B.6: Comparison of different methods to apply attention on SBD-$5^i$.

| Metric | Method (5-shot) | SBD-$5^0$ | SBD-$5^1$ | SBD-$5^2$ | SBD-$5^3$ | Mean |
|---|---|---|---|---|---|---|
| MF (ODS) | Attentive Decoding | **34.71** | **36.81** | **32.02** | **28.37** | **32.98** |
| | Direct Attention | 33.47 | 36.64 | 31.85 | 28.28 | 32.56 |
| | No Attention | 31.75 | 33.41 | 28.44 | 26.03 | 29.91 |
| AP | Attentive Decoding | **30.47** | **32.40** | **27.01** | **23.06** | **28.24** |
| | Direct Attention | 30.31 | 31.93 | 26.15 | 22.59 | 27.75 |
| | No Attention | 27.61 | 28.39 | 22.66 | 20.11 | 24.69 |
| Metric | Method (1-shot) | SBD-$5^0$ | SBD-$5^1$ | SBD-$5^2$ | SBD-$5^3$ | Mean |
| MF (ODS) | Attentive Decoding | **31.54** | **34.75** | 29.47 | **26.68** | **30.61** |
| | Direct Attention | 31.27 | 34.53 | 30.01 | 26.41 | 30.56 |
| | No Attention | 30.72 | 33.03 | 28.63 | 25.04 | 29.36 |
| AP | Attentive Decoding | **26.81** | **29.08** | **23.77** | **20.44** | **25.03** |
| | Direct Attention | 26.46 | 28.61 | 23.64 | 20.23 | 24.74 |
| | No Attention | 26.10 | 27.21 | 22.47 | 18.81 | 23.65 |

Table B.7: Comparison of different methods to generate attention on SBD-$5^i$.

| Metric | Method (5-shot) | SBD-$5^0$ | SBD-$5^1$ | SBD-$5^2$ | SBD-$5^3$ | Mean |
|---|---|---|---|---|---|---|
| MF (ODS) | Segmentation Attention | **34.71** | 36.81 | **32.02** | **28.37** | **32.98** |
| | Edge Attention | 33.83 | **37.01** | 31.14 | 27.92 | 32.48 |
| AP | Segmentation Attention | **30.47** | **32.40** | **27.01** | **23.06** | **28.24** |
| | Edge Attention | 29.16 | 32.35 | 25.56 | 22.52 | 27.40 |
| Metric | Method (1-shot) | SBD-$5^0$ | SBD-$5^1$ | SBD-$5^2$ | SBD-$5^3$ | Mean |
| MF (ODS) | Segmentation Attention | **31.54** | **34.75** | 29.47 | **26.68** | **30.61** |
| | Edge Attention | 30.25 | 33.58 | 30.64 | 26.16 | 30.16 |
| AP | Segmentation Attention | **26.81** | **29.08** | 23.77 | **20.44** | **25.03** |
| | Edge Attention | 25.21 | 27.83 | 24.87 | 20.34 | 24.56 |

# C   QUALITATIVE RESULTS

In Figures C.1 and C.2, we visualize the qualitative results of CAFENet for FSE-1000 and SBD-$5^i$, respectively. We can see that the proposed CAFENet successfully detect the edge of the target class from given images. In Figure C.1, we can compare the qualitative results of edge predictions from different methods. We can see that the 'DFF + Finetune' method succeeds in finding the edges of the objects, but it lacks the ability to learn semantic information and hardly distinguish the boundary of target class from the other boundaries. For 'PANet + Sobel' method, on the other hand, it successfully understands the semantic information and localizes the target object, but it fails to refine the correct boundary. The proposed CAFENet, however, is capable of localizing the objects from target class and detecting the correct boundary at the same time.

Figure C.3 visualizes more qualitative results on SBD-$5^i$ from ablation experiments. We illustrate and compare the boundary prediction results of the baseline, **Seg**, **Seg + Att**, and **Seg + Att + MSMR** methods. For fair comparison, all methods share the same support set. From the results, we can clearly see that the techniques proposed in CAFENet steadily improve the quality of edge prediction.

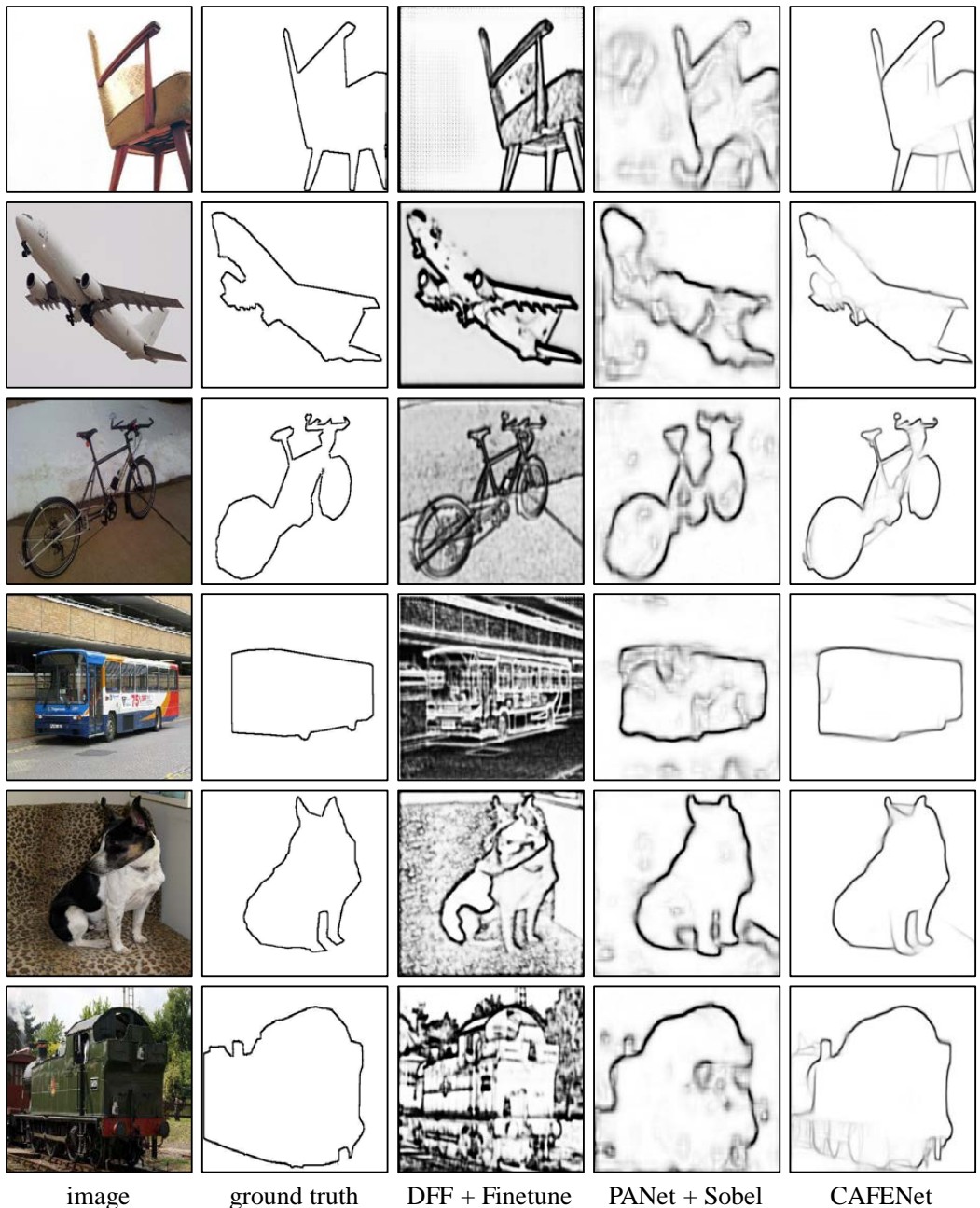

image     ground truth     DFF + Finetune     PANet + Sobel     CAFENet

Figure C.1: Qualitative examples of 5-shot edge detection on SBD-$5^i$ dataset.

# D LABEL GENERATOR

## D.1 EDGE LABEL GENERATOR

Algorithm D.1 generates the edge labels from the segmentation labels. The edge label generator finds the regions where the pixel value of segmentation label drastically changes, and determine the pixels in the regions as the boundary. Note that the pixels at the border of the image are also determined as the boundary.

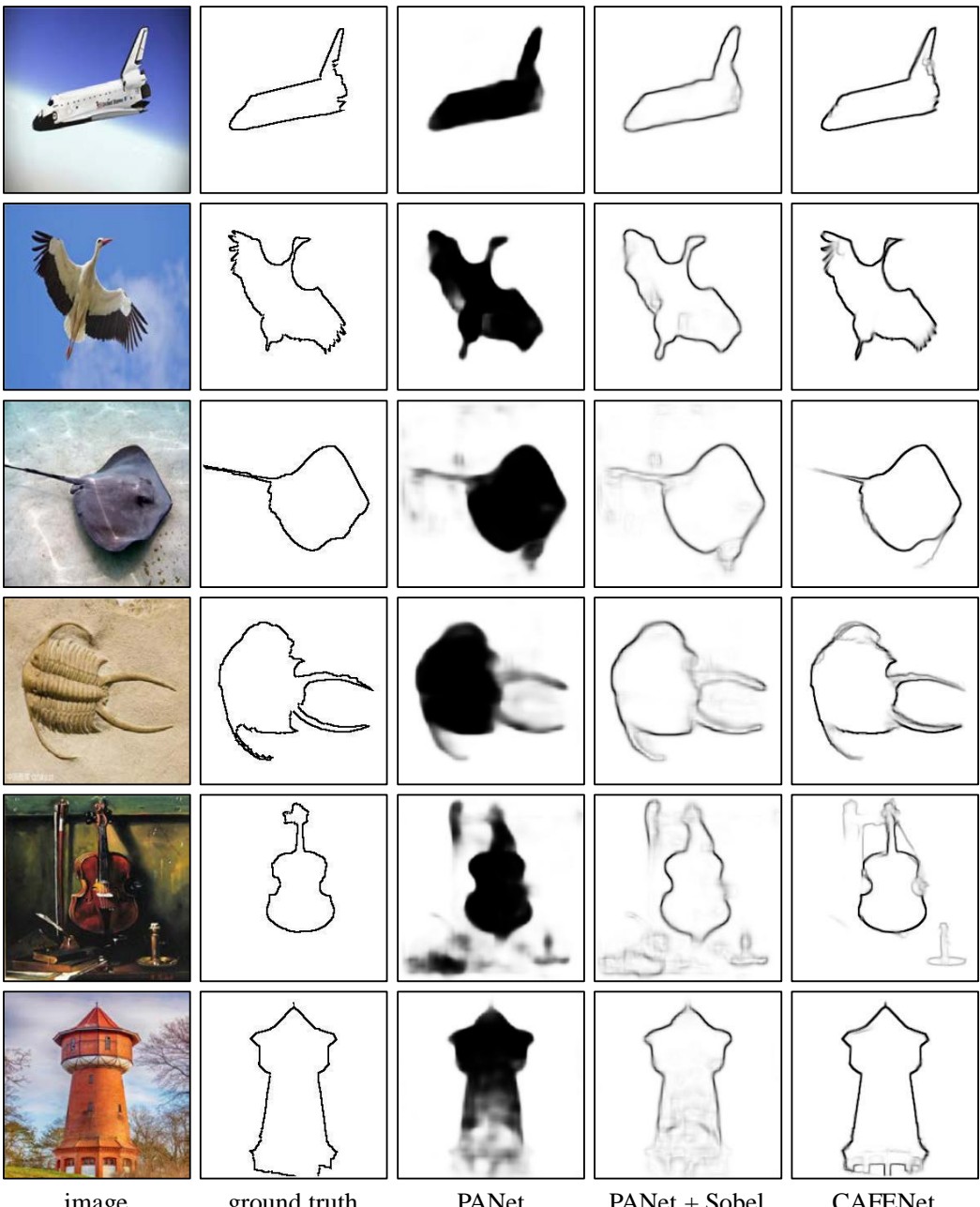

| image | ground truth | PANet | PANet + Sobel | CAFENet |

Figure C.2: Qualitative examples of 5-shot edge detection on FSE-1000 dataset.

## D.2 SEGMENTATION LABEL GENERATOR

Algorithm D.2 is the segmentation label generator which generates the segmentation label from the edge label. Before the label generation, the pixels are divided into several groups based on boundary labels. We employ the Breadth-First Search (BFS) algorithm and divide pixels into groups $\{G^1, G^2, ..., G^n\}$. The segmentation label generator of Algorithm D.2 classifies these groups into foreground and background. First, the algorithm sweeps each column and row to count the number of pixel value change in edge label. If there are certain numbers of changes, the algorithm again sweeps the column or row and record the location of foreground pixels and mark the foreground pixels in the column or row in a matrix $T$. Based on pixel groups $\{G^1, G^2, ..., G^n\}$, the marking results in $T$ are then divided into pixel value groups $\{T^1, T^2, ..., T^n\}$. The probability that each

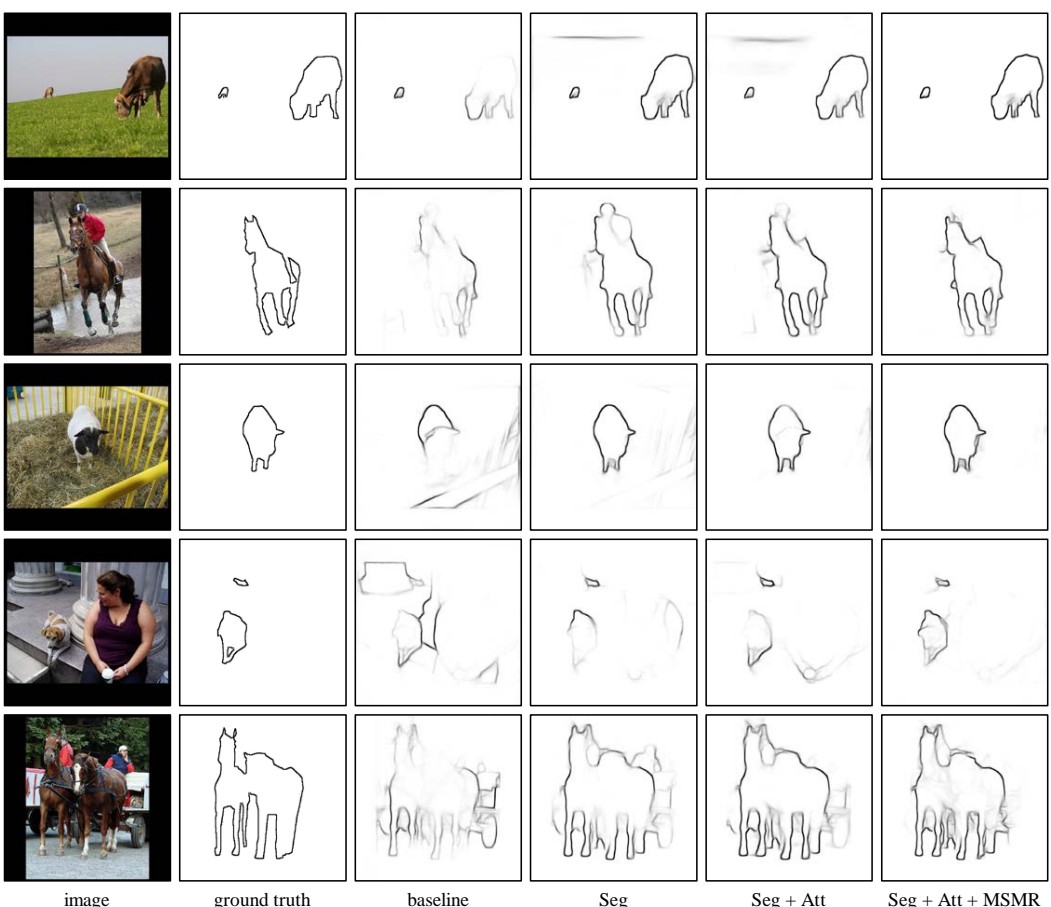

image        ground truth        baseline        Seg        Seg + Att        Seg + Att + MSMR

Figure C.3: Qualitative results comparison of ablation experiments

---

**Algorithm D.1** Edge Label Generation

---

**Input:** Segmentation label $M$ of an image
**Output:** Edge label $y$ of an image.

    $y \leftarrow 0_{W \times H}$                          ▷ Initialize $y$ as zero matrix having same shape with $M$
    **for** $(h, w)$ in $(1, 1),...,(H,W)$ **do**                 ▷ $H/W$ is height/width of the image
        **if** $M_{h,w} = 1$ **then**
            **for** $(a, b) = (-r,-r),...,(r,r)$ **do**        ▷ radius $r$ determines thickness of edge
                **if** $M_{h+a,w+b} = 0$ **then**
                    $y_{h,w} \leftarrow 1$            ▷ 0/1 means non-edge/edge pixel, respectively
                    **break**
                **else if** $(h + a < 0)$ or $(h + a > H)$ or $(w + b < 0)$ or $(w + b > W)$ **then**
                    $y_{h,w} \leftarrow 1$
                    **break**
    **return** $y$                                 ▷ Return label annotation

---

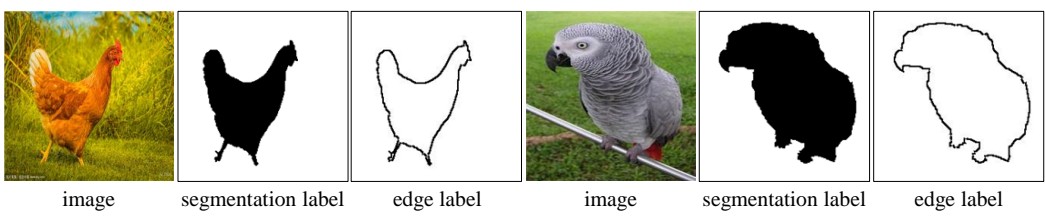

image        segmentation label        edge label        image        segmentation label        edge label

Figure D.1: Result of edge label generator.

group $G^i$ belongs to the foreground is calculated as the mean of pixel values $T^i$. The groups with probability above the threshold $\lambda$ are determined as the foreground groups, and the pixels belongs to foreground groups are marked as foreground pixels. We set the threshold value $\lambda$ to be $20/255$.

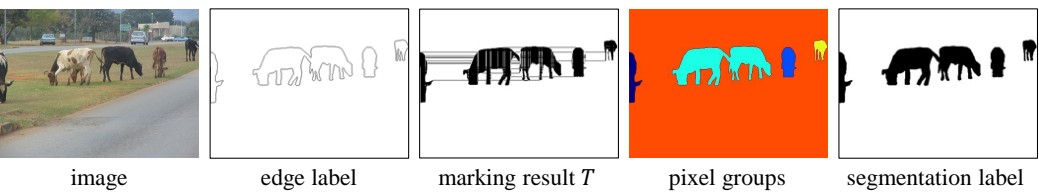

image        edge label        marking result $T$        pixel groups        segmentation label

Figure D.2: Result of segmentation label generator.

# E  DETAILS ON DATASETS

## E.1  FSE-1000

We build FSE-1000 using an existing few-shot segmentation dataset, FSS-1000. We extract the boundary labels from segmentation annotations using Algorithm D.1. The radius value $r$ in Algorithm D.1 is set to 3 in FSE-1000. 1000 categories in FSE-1000 are split into 800 train classes and 200 test classes. For the detailed class configuration, the reader may refer to our attachment on class configuration. Figure D.1 visualizes the result of our edge extraction algorithm.

## E.2  SBD-$5^i$

SBD-$5^i$ is constructed based on the existing semantic edge detection dataset (SBD). Due to the noise of boundary annotations in original SBD, we utilize the thicker edge as done in FSE-1000. To extract thicker edge, we generate the segmentation labels from the edge labels using Algorithm D.2 instead of using existing segmentation labels of SBD. Figure D.2 shows the process of generating the segmentation label from the edge label. From the generated segmentation labels, we extract edge

---

**Algorithm D.2** Segmentation Label Generation

---

**Input:** Edge label $y$ of an image, pixel groups $G^1, G^2, ..., G^n$
**Output:** Segmentation label $M$ of an image.

$\quad M, T \leftarrow 0_{W,H}$         ▷ Initialize $M, T$ as zero matrix having same shape with $y$
$\quad$ **for** $h = 1,...,H$ **do**         ▷ $H$ is height of the image
$\quad\quad cnt, mode \leftarrow 0$
$\quad\quad$ **for** $w = 1,...,W$ **do**         ▷ $W$ is width of the image
$\quad\quad\quad$ **if** $y_{h,w} = mod(mode + 1, 2)$ **then**      ▷ Accumulate changes of pixel value
$\quad\quad\quad\quad cnt \leftarrow cnt + 1$
$\quad\quad\quad\quad mode \leftarrow mod(mode + 1, 2)$
$\quad\quad$ **if** $mod(cnt, 4) = 0$ and $cnt \neq 0$ **then**     ▷ Check if there are FG pixels in the row
$\quad\quad\quad cnt', mode' \leftarrow 0$
$\quad\quad\quad$ **for** $w' = 1,...,W$ **do**        ▷ Find location of FG pixels in the row
$\quad\quad\quad\quad$ **if** $y_{h,w'} = mod(mode' + 1, 2)$ **then**
$\quad\quad\quad\quad\quad cnt' \leftarrow cnt' + 1$
$\quad\quad\quad\quad\quad mode' \leftarrow mod(mode' + 1, 2)$
$\quad\quad\quad\quad$ **if** $mod(cnt', 4) = 2$ **then**
$\quad\quad\quad\quad\quad T_{h,w'} \leftarrow 1$        ▷ Record location of FG pixels in the row
$\quad$ **for** $w = 1,...,W$ **do**        ▷ Repeat the same process for every column
$\quad\quad cnt, mode \leftarrow 0$
$\quad\quad$ **for** $h = 1,...,H$ **do**
$\quad\quad\quad$ **if** $y_{h,w} = mod(mode + 1, 2)$ **then**
$\quad\quad\quad\quad cnt \leftarrow cnt + 1$
$\quad\quad\quad\quad mode \leftarrow mod(mode + 1, 2)$
$\quad\quad$ **if** $mod(cnt, 4) = 0$ and $cnt \neq 0$ **then**
$\quad\quad\quad cnt', mode' \leftarrow 0$
$\quad\quad\quad$ **for** $h' = 1,...,H$ **do**
$\quad\quad\quad\quad$ **if** $y_{h',w} = mod(mode' + 1, 2)$ **then**
$\quad\quad\quad\quad\quad cnt' \leftarrow cnt' + 1$
$\quad\quad\quad\quad\quad mode' \leftarrow mod(mode' + 1, 2)$
$\quad\quad\quad\quad$ **if** $mod(cnt', 4) = 2$ **then**
$\quad\quad\quad\quad\quad T_{h',w} \leftarrow 1$
$\quad$ **for** $i = 1, ..., n$ **do**
$\quad\quad T^i \leftarrow T_{h,w|(h,w) \in G^i}$
$\quad\quad$ **if** $mean(T^i) \geq \lambda$ **then**      ▷ Check the probability that $G^i$ belongs to foreground
$\quad\quad\quad M_{h,w|(h,w) \in G^i} \leftarrow 1$        ▷ 1 means a foreground pixel
$\quad\quad$ **else**
$\quad\quad\quad M_{h,w|(h,w) \in G^i} \leftarrow 0$        ▷ 0 means a background pixel
$\quad$ **return** $M$        ▷ Return segmentation annotation

---

labels using Algorithm D.1 with a radius value of 4. This process allows us to train the proposed CAFENet using only the edge labels.

While all images in FSE-1000 have the same size, images in SBD-$5^i$ have different size. However, constructing the training episode as a mini-batch requires images with the same size. Previous works on semantic edge detection typically apply random cropping to deal with this issue. For the few-shot setting, however, random cropping severely degrades informativeness of support set and consequently hinders learning. Alternatively, we utilize the training examples resized to $320 \times 320$ to maintain the information of images as much as possible. When resizing the edge labels for training, we first generate segmentation labels in original scale using Algorithm D.2 and resize the segmentation labels to $320 \times 320$. Then, we extract edge labels from resized segmentation labels using Algorithm D.1 with a radius value of 3.

# F  ADDITIONAL RESULTS WITH MULTI-ANGLE INPUT TEST

In this section, we report the few-shot semantic edge prediction results with multi-angle input test. In multi-angle input test, the model predicts the edge by averaging the 4 edge prediction results from 4 copies of an input image rotated by multiples of 90 degrees. We have empirically found that multi-angle input test significantly improves the performance. Table F.1 and F.2 show the evaluation results with multi-angle input test for FSE-1000 and SBD-$5^i$, respectively. We can verify the effectiveness of the multi-angle input test from the results.

Table F.1: 1-way 5-shot results with the multi-angle input test on FSE-1000. 1000 randomly sampled test episodes are used for evaluation. MF and AP scores are measured by %

| Metric | Method | 1-way 5-shot |
|---|---|---|
| MF (ODS) | baseline | 55.23 |
| | Seg | 61.17 |
| | Seg + Att | 61.90 |
| | Seg + Att + MSMR | **62.23** |
| AP | baseline | 56.12 |
| | Seg | 63.32 |
| | Seg + Att | 63.83 |
| | Seg + Att + MSMR | **65.81** |

Table F.2: 1-way 5-shot results with the multi-angle input test on SBD-$5^i$. 1000 randomly sampled test episodes are used for evaluation. MF and AP scores are measured by %

| Metric | Method(5-shot) | SBD-$5^0$ | SBD-$5^1$ | SBD-$5^2$ | SBD-$5^3$ | Mean |
|---|---|---|---|---|---|---|
| MF (ODS) | baseline | 24.38 | 23.78 | 22.81 | 20.39 | 22.84 |
| | Seg | 32.46 | 33.18 | 29.63 | 26.46 | 30.43 |
| | Seg + Att | 34.29 | 35.76 | 32.25 | 28.12 | 32.61 |
| | Seg + Att + MSMR | **36.13** | **37.92** | **34.18** | **30.21** | **34.61** |
| AP | baseline | 21.03 | 20.39 | 17.48 | 16.31 | 18.80 |
| | Seg | 27.92 | 28.00 | 23.94 | 20.06 | 24.98 |
| | Seg + Att | 29.94 | 30.40 | 24.34 | 21.38 | 26.52 |
| | Seg + Att + MSMR | **31.92** | **33.51** | **29.28** | **24.91** | **29.91** |

