# OpenReview forum: "CAFENet: Class-Agnostic Few-Shot Edge Detection Network"
_ICLR.cc/2021/Conference — Reject_

### Official Review · AnonReviewer1 · 2020-10-28
**New task of Few-Shot Edge Detection along with a proposed method CAFENet**

**Rating:** 4
**Confidence:** 5

**Review:**

This paper proposes the few-shot edge detection task, which is similar to few-shot segmentation but for the dual task of detecting semantic edges. For the task, the authors construct datasets and experimental settings constructed from existing edge detection dataset (BSD) and a few shot segmentation dataset (FSS). For the proposed method, the authors:

1) Propose a meta-learning strategy based on prototypical learning and using intermediate low-res segmentation prediction to generate attention maps for few shot edge detection
2) Propose using a multi-split matching which performs prototypical matching (for fg and bg) using many splits of the high dimensional feature

This paper is the first to propose the task of few shot semantic edge detection (dual task of few shot segmentation, which has been explored before). This is a relevant problem, and is well suited for few shot tasks instead of segmentation since learning a boundary can be easier (modelling high frequency info) as compared to modelling interior pixels (often containing low frequency info) that has to be done in few shot segmentation. The method section is also clearly written.

I have **two major issues** with the paper in its current form:

1. Results of PANet + Sobel filter:
- The dataset constructed in this paper thicken edges by a certain radius (as explained in the paper and in the supplementary material) to reduce the time complexity of computing ODS. Quote - "Moreover, evaluation with non-zero distance tolerance requires additional heavy computation. This becomes more problematic under few-shot setting where the performance should be measured on the same test image multiple times due to the variation in the support set".
- However, PANet + Sobel filter would only generate thin edges and there is no mention in the paper of the authors thickening its output edges according to the dataset (whereas thick edges are used to train CAFENet, so it has learned to produce thicker edges). Quote - "For each split of SBD-5i, we meta-train the PANet on training classes using the segmentation labels. In evaluation, we obtain the edge predictions of test classes by applying the Sobel edge detector on the segmentation masks as done in (Acuna et al., 2019), and compare the predictions with the edge labels of test classes"

This evaluation mismatch is a serious flaw and I would love to either get an explanation for why this is actually correct or updated results on these.

Generally, my view on this is that the evaluation should follow previous methods and should be done only on *exact* ground truth edges. Since exact ground truth edges are unavailable, using a error threshold makes sense. Evaluating with thicker edges as ground truth unfairly up-scores methods that generate thicker edges themselves, which are definitely not desirable. Therefore, I don't think the dataset with thickened edges as proposed is viable to benchmark improvements in the future for few-shot edge detection.

2. Multi-Split Matching Regularization - The proposed MSMR technique is well ablated for the different choices of its hyperparameters, but at the end of the day is still an ad-hoc technique. What other design choices could be used? The proposed splitting of the vectors into smaller vectors is basically a fixed set of projections from the original feature space to a lower dimension where the different projections discard a lot of dimensions. Is this the best choice? There is a lot of theory behind random projections -- how would they work here? Instead of using K splits into N/K size, would K random projections into N/K size perform better? I believe this needs to be ablated more, and MSMR needs to be grounded more in existing work.

Overall, I believe 1. is a serious flaw and 2. is something that would make the paper much more attractive to a reader and provide more insight into why MSMR results in additional improvements. I am rating this paper at a 4 now and hoping that my assessment of issue 1 is a mistake.

---

> ### Author Response · Authors · 2020-11-17
> **Response to Reviewer 1**
>
> Response to concerns about PANet+Sobel baseline: Indeed, we did measured the perfornance of PANet+Sobel on various sobel filter size. As reviewer mentioned, we first set sobel filter size to 1 and observed unsatisfactory result. We empirically found that optimal kernel size is 3 and conduct comparision accordingly. If given a chance, we will elaborate on detail in experimental setting. Moreover, as shown in Figure C.2 and Figure C.1 in appendix, PANet+Sobel indeed generates sufficiently thick edge prediction compared to that of CAFENet (Ours)
>
> Response to concerns about evaluation setting: We believe our answer can be summarized as follows: First, our thicked edge label is not too thick to be unfair. In SBD-5i, the edge we thicken is 3 to 4 pixels in images of 512x512 pixel size, which is not that big compared to the existing edge width (1 to 2 pixels) in SBD dataset. In evaluation of most edge detection work, a value of tolerance is used to alleviate noise in edge label. In CASEnet [1] and SEAL [2] , a tolerance of 2% is given to the SBD dataset, which means that an error of about 10 pixels is allowed in a 512x512 image. We believe that it is more accurate to measure the performance without tolerance on a relatively less noisy edge label than to give high tolerance on an inaccurate thin edge label. Also, inaccurate noisy labels have an adverse effect on training edge detector networks. STEAL used active alignment to create a more accurate edge label and trained an edge detector using that aligned label to overcome the infamous noise label of the SBD dataset.  Therefore, we expect that the noise-reduced thickened edge label can also help train an accurate edge detector.
> Finally, we believe that our proposed dataset is viable enough to improve performance. CAFENet showed qualitative results for relatively easy categories such as cat and cow, but it was observed that even localization (semantic segmentation) was often performed improperly for difficult categories such as plant, tv, and table. Of course, it is important for the semantic edge detector to generate thin and crips boundaries, but the ability to grasp semantic information and localize objects seems to be dominate in few-shot segmentation. You can confirm this by significant perforance gap between FSE-1000 (Table.2) and SBD-5i (Table.1). Edge widths of two datasets are similar, but the perfomance severely degrades in SBD-5i since it contains more cluttered and difficult images. Therefore, we believe that there is ample room left for further work to improve the performance.
>
> [1] Yu, Z., Feng, C., Liu, M. Y., & Ramalingam, S. (2017). Casenet: Deep category-aware semantic edge detection. In Proceedings of the IEEE Conference on Computer Vision and Pattern Recognition (pp. 5964-5973).
>
> [2] Yu, Z., Liu, W., Zou, Y., Feng, C., Ramalingam, S., Vijaya Kumar, B. V. K., & Kautz, J. (2018). Simultaneous edge alignment and learning. In Proceedings of the European Conference on Computer Vision (ECCV) (pp. 388-404).
>
> Response to concerns about MSMR: We conducted additional ablation experiments on MSMR with random projections (see Appendix.B.1.3 for more information). Experiment results show that applying MSMR with K random projections with N/K size harms the performance of proposed CAFENet. Actually, the performance has been degraded to about two-thirds in the experiments with SBD-5i dataset. Concretely, average precision(AP) has dropped from 28.24 to 18.07 in 5-shot setting, and from 25.03 to 17.92 in 1-shot setting. Maximum F-measure(MF) has dropped from 32.98 to 20.64 in 5-shot setting and from 30.61 to 20.58 in 1-shot setting.

---

### Official Review · AnonReviewer3 · 2020-10-29
**a problem of few-shot semantic edge detection introduced**

**Rating:** 6
**Confidence:** 3

**Review:**

Summary:
This paper introduces a novel problem of "few-shot semantic edge detection" where semantic boundaries are to be learned/detected with a few labeled samples. In order to remedy the issue of label sparsity within the few-shot scenario, the authors have leveraged the use of the segmentation process which provides the semantic information to the edge detector. They have also incorporated a meta-learning approach, namely Multi-Split Matching Regularization (MSMR), to avoid the overfitting when high-dimensional embeddings are used for feature matching.

Pros:
- The motivation was effectively described and the problem was well-defined which made it easy to read.
- Proposed architecture smartly inherits many previously introduced components (e.g., prototype-based metric learner)and combines them into a well-working framework to train a few-shot edge detector.
- Quantitative results show reasonable edge detection performance

Cons/Questions:
- The authors constructed the baseline by combining PANet (ICCV'19) with the Sobel edge detector while disregarding some of the better/recent approaches shown below:
*  Canet: Class-agnostic segmentation networks with iterative refinement and attentive few-shot learning. (CVPR'19)
** Prototype Mixture Models for Few-shot Semantic Segmentation (ECCV '20)
*** Part-aware Prototype Network for Few-shot Semantic Segmentation (ECCV '20)
Would the proposed approach still be effective than the combined version of Segmentators+Sobel if the segmentors were updated to more recent models?

- Where does the proposed approach fit w.r.t the previous approaches mentioned in Section 2.2 and 2.3?
- Theoretical or analytical reasoning to support the effectiveness of MSMR would better convey the authors' claim

Minor comments:
- There might be some incorrect labels in Figure 3. In this figure, the diagram on the left is not supposed to be split-wise matching, right?

---

> ### Author Response · Authors · 2020-11-17
> **Response to Reviewer 3**
>
> Response to concerns about baselines: We appreciate the comment; we conducted experiments using the more advanced few-shot segmentation method of Prototype Mixture Model, PMM [1]. We applied the Sobel edge detector to the output of PMM, and the experiment results show that the proposed CAFENet still works better than the combination of few-shot segmentator and edge detector, even when we utilize more advanced few-shot segmentator equipped with a bigger backbone(ResNet-50) that ours(ResNet-34)
>
> [1] Yang, B., Liu, C., Li, B., Jiao, J., & Ye, Q. (2020). Prototype Mixture Models for Few-shot Semantic Segmentation. arXiv preprint arXiv:2008.03898.
>
> Response to question about comparison with previous approaches: The proposed CAFENet is actually an effective combination of semantic edge detection of Section 2.2 and few-shot segmentation of Section 2.3 (or few-shot learning of Section 2.1). Existing semantic edge detection networks are trained only using edge mask. However, in the few-shot learning scenario, it is difficult to obtain semantic information only with edge labels because the edge label itself represents only a very limited part of the target image. To solve this problem, we preprocess the edge label to create a segmentation label (See section D, E in appendix for more information) and perform perform few-shot segmentation to support the few-shot edge detection process to get high performance. To validate our approach, we train our network only using edge masks to produce attention map(“baseline” in Section C of appendix), and it significantly drops the performance.
>
> Response to minor comments: The digram on the left is not supposed to be split-wise matching. It should be pixel-wise matching, and we fix the diagram and uploaded the manuscripted with correct diagram. Thank you for your careful observation.

---

### Official Review · AnonReviewer2 · 2020-10-29
**This work poses a new few-shot learning task, i.e., few-shot semantic edge detection, and proposes a Class-Agnostic Few-shot Edge detection Network to tackle this problem. In addition, two new datasets, FSE-1000 and SBD-5i, were introduced to evaluate the performance.**

**Rating:** 4
**Confidence:** 5

**Review:**

Pros:
1 This work is well written and easy to understand.
2 Extensive ablation experiments are performed to verify the effectiveness of the proposed modules in the CAFENet.

Cons:
1 This new task is very similar to the few-shot segmentation. What are the advantages of this task over few-shot segmentation? The significance of this new research is doubtful.

2 The proposed method was named “class-agnostic”. However,  it seems the basic setting is exactly the same as few-shot segmentation, i.e., if I want to segment the horse in the given query, I need to provide a hose image and its mask. So, I think the method is class-aware rather than class-agnostic.

3 In my opinion, the main technical contribution of this work lies in the split-wise matching, which is employed to produce better segmentation masks. It is hard to find novel/specific designs in the edge detector part. So, I consider that the performance will significantly rely on the predicted segmentation masks, making this new task back to few-shot segmentation.

4 Using PANet + Sobel as a baseline is unfair. Authors should replace the few-shot segmentation branch with PANet and more latest advanced few-shot segmentation methods for comparison.

---

> ### Author Response · Authors · 2020-11-17
> **Response to Reviewer 2**
>
> Response to Cons 1: We stress few-shot edge detection is different from few-shot segmentation, althouth they bare some similarity. The few-shot edge detection model works much better than the few-shot segmentation model when finding the boundary of the object, and we show this with various experiments. In Section 5.4 and Appendix C, we confirm that the combination of few-shot segmentator (PANet) and Sobel edge detector generates inaccurate and blur edge prediction despite of its fine localization performance, while the proposed CAFENet generates more accurate and clear edge prediction results.
>
> Response to Cons 2: We use the keyword of “class-agnostic” in the sense that our edge detector is not limited to a specific seen classes, but performs well for an arbitrary unseen class. A similar example of the keyword “class-agnostic” can be found in the title of [1], which performs few-shot semantic segmentation.
>
> [1] Zhang, C., Lin, G., Liu, F., Yao, R., & Shen, C. (2019). Canet: Class-agnostic segmentation networks with iterative refinement and attentive few-shot learning. In Proceedings of the IEEE Conference on Computer Vision and Pattern Recognition (pp. 5217-5226).
>
> Response to Cons 3: We note that applying the segmentation module is one of our main contributions. No one had tried to exploit segmentation modules to support a semantic edge detector. This work is first to propose how to extract segmentation labels from edge GT and employ them (see Supplementary Material for more details on the segmentation label extractor). The performance leap from baseline to segmentation module is the largest, and we believe this is also our key contribution. In addition, the gains from attention and MSMR are meaningful, depending on the test set. For example, for 1 way 5-shot result of Table B.1 in Appendix, on the AP metric, “Seg+Att+MSMR” yields 63.92 versus 61.87 for “Seg” only, which is significant. Furthermore, one clearly sees qualitative improvements in the prediction of sematic edges in Fig. C.3 of the Appendix.
> Comparing the images of a person riding a horse between “Seg” and “Seg+Att+MSMR”, one clearly sees only the edge of the horse in “Seg+Att+MSMR” as desired, whereas both the human and the horse are still identified in “Seg”.
>
> Response to Cons 4: We agree that comparing our CAFENet with the more advanced few-shot segmentation method would show the superiority of our method more clearly. We appreciate your valuable comment, and we conducted the experiments using more advanced few-shot segmentation method of Prototype Mixture Model, PMM [2]. We applied the Sobel edge detector to the output of PMM, and the experiment results show that the proposed CAFENet still works better than the combination of few-shot segmentator and edge detector, even when we utilize more advanced few-shot segmentator equipped with a bigger backbone(ResNet-50) that ours(ResNet-34)
>
> [2] Yang, B., Liu, C., Li, B., Jiao, J., & Ye, Q. (2020). Prototype Mixture Models for Few-shot Semantic Segmentation. arXiv preprint arXiv:2008.03898.

---

### Official Review · AnonReviewer4 · 2020-10-30
**A straghitfoward combination of few shot segmentation+edge detection**

**Rating:** 4
**Confidence:** 5

**Review:**


A. Summary:
This paper works on few-shot semantic edge detection. Instead of dealing with the problem in a single stage, the authors decompose the problem into two stages.  First, a few-shot segmentation stage, where the foreground and the background probability are estimated via attention with the foreground and the background prototype (averaged feature vector on the foreground and the background region). Second, the feature maps from the encoder are masked by the attention map and sent to the decoder to generate the final edge-map.
In the experiments, the authors modify two existing datasets for evaluation.

B. Strength:
1.  The method is clearly described and easy to reproduce.
2. Enforcing the group-wise similarity in sec.4.2 is a reasonable regularization.
3. The two-stage decomposition separates the high-level sub-task (semantics) from the low-level sub-task (edge), which makes the training easier.

C. Weakness:
1. The attention map preserves the internal region of a class. However, this paper is about semantic edge detection, where the internal region does not contain any edge. So the author can try to process the final attention map by distance transform, which only focus on the region around the edge of the attention map and potentially further improve the results.
2. The comparison with previous methods, especially the PANet+Sobel is unfair. The semantic response map corresponds to the attention map in the proposed method, so the author should also use sobel operator on their attention map to have a fair comparison, which can also serve as an ablation study to verify whether the attention map is one of the major contributions to the edge quality.
3. Another ablation study worth to be investigated is to apply the attention map directly on the edge map, which will further sperate the high-level sub-task(semantic attention maps) from the low-level sub-task  (edge-detection).

Some minor issues:
1.  The attention map in Fig.2 (positive is black)is inconsistent with the rest of the paper (positive is white).

D. Justification of the score:
I vote for the current score because this paper lacks sufficient insights and novelty both algorithmically and conceptually. It also contains some issues in evaluation.

E. Expectation for the rebuttal:
I hope the authors could address my questions in C.weakness.

---

> ### Author Response · Authors · 2020-11-17
> **Response to Reviewer 4**
>
> Response to weakness 1: We appericiate the comment; we have conducted experiments with the attention map that makes decoder to focus on the region around the edge of the semantic response map (segmentation module output). Specifically, we generated the edge attention map by applying an average pooling on segmentation module output and subtracted the result from segmentation module output, following [1]. Experiment results show that applying edge attention map actually degrades the performance a bit (less than 1% for both AP and MF). We added the result of this ablation study in Appendix.
>
> [1] Feng, M., Lu, H., & Ding, E. (2019). Attentive feedback network for boundary-aware salient object detection. In Proceedings of the IEEE Conference on Computer Vision and Pattern Recognition (pp. 1623-1632).
>
> Response to weakness 2: We would like to note that the goal of the comparison with PANet + Sobel baseline is to show that it is impossible for a combination of few-shot segmentator and edge detector to replace few-shot edge detector. So as [2] did, we compared edge predictions obtained by applying Sobel operator to the output of PANET with the edge predictions from proposed CAFENet.
>
> [2] Acuna, D., Kar, A., & Fidler, S. (2019). Devil is in the edges: Learning semantic boundaries from noisy annotations. In Proceedings of the IEEE Conference on Computer Vision and Pattern Recognition (pp. 11075-11083).
>
> However, we agree that applying sobel operator on our attention map would be an interesting ablation study. As the reviewer suggested, we applied sobel operator to up-scaled segmentation output and compared it (Seg+Sobel) with PANET+Sobel. As a result, Seg+Sobel shows much inferior performance than PANet+Sobel. Specifically, Seg+Sobel has 2.09% lower mean AP and 6.79% mean MF in 5-shot setting. Also, Seg+Sobel has 3.41% lower mean AP and 8.04% Mean MF in 1-shot setting. The reason behind this result is as follows: The size of our segmentation module output is only 1/16 of the input image. It is impossible to directly compare the performance since the role of our segmentation module is localizing target objects and extracting semantic knowledge.
>
> Response to weakness 3: We again agree that this is also an interesting ablation study. We conducted another experiment of applying the attention map directly to the edge map, without attentive decoding. We recorded the experimental result in Appendix B.2.
> Applying attention map to the edge map generates better performance than non-attention method. Specifically, average precision(AP) increased by 3.66% in 5-shot setting and 1.09% in 1-shot setting. Maximum F-measure(MF) increased by 2.65% in 5-shot setting and 1.20% in 1-shot setting. This results show the effectiveness of semantic encoding. The experiments also show that our approach (applying attention map before edge detector) yields better results.
>
> Response to minor issues: We revised the Figure 2 and some other figures in our updated manuscript to make style of figures consistent.

---

### Decision · Program_Chairs · 2021-01-07
**Final Decision**

**Decision:**

Reject

**Comment:**

The paper introduces the new task of few-shot semantic edge detection by adapting existing datasets. It proposes a new method which is compared to a baseline.

Pros:
- Clear writing.
- Extensive ablation experiments.
- Good architectural choices.

Mixed:
- The value of the new task raises a mix of opinions. For example R1 sees it as a "relevant problem, and is well suited for few shot tasks", but R2 finds is very similar to few-shot segmentation. I think a more interesting version of the problem (that would also create more separation to few-shot segmentation) would be to also consider internal edges, not just "semantic boundaries". For example the original BSDS dataset has pure edge annotations.
- Besides the task, another novelty of the paper is the proposed multi-split matching technique, but while it is well demonstrated empirically (as backed by additional results given by authors in rebuttal), R3 would like to have seen "theoretical or analytical reasoning" and R1 says it is an "ad-hoc technique".

Cons:
- the PANet+Sobel baseline. All 4 reviewers are unhappy with this baseline: 3 of them find it unfair because of the non-standard edge thickening employed and 2 think there would be more recent and better baselines. The authors provided a rebuttal arguing that their GT edges are "not too thick to be unfair" but two of the reviewers mentioned they remained unconvinced -- R1 hopes "the authors will work on cleaner evaluation of the baseline" and R4 find the baseline "still unconvincing in the revised version".

Overall the paper would benefit from one more iteration focusing on the evaluation procedure to be convincing and impactful.